# Quantifying the effect of interpregnancy maternal weight and smoking status changes on childhood overweight and obesity in a UK population-based cohort

**Elizabeth J. Taylor**[1,2,3], **Nida Ziauddeen**[3,4], **Ann Berrington**[5], **Keith M. Godfrey**[1,6], **Nisreen A. Alwan**[3,4,7] *

**1** NIHR Southampton Biomedical Research Centre, University of Southampton and University Hospital Southampton NHS Foundation Trust, Southampton, United Kingdom, **2** Nuffield Department of Population Health, Big Data Institute, University of Oxford, Oxford, United Kingdom, **3** School of Primary Care, Population Sciences and Medical Education, Faculty of Medicine, University of Southampton, Southampton, United Kingdom, **4** NIHR Applied Research Collaboration Wessex, Southampton, United Kingdom, **5** Department of Social Statistics and Demography, University of Southampton, Southampton, United Kingdom, **6** MRC Lifecourse Epidemiology Centre, University of Southampton, Southampton General Hospital, Southampton, United Kingdom, **7** University Hospital Southampton NHS Foundation Trust, Southampton, United Kingdom

* N.A.Alwan@soton.ac.uk

**Data Availability Statement:** The study's ethical approval from the Faculty of Medicine Ethics Committee, University of Southampton and the

## Abstract

### Background

Maternal preconception and pregnancy exposures have been linked to offspring adiposity. We aimed to quantify the effect of changes in maternal weight and smoking status between pregnancies on childhood overweight/obesity ($\geq 85^{\text{th}}$ centile) and obesity ($\geq 95^{\text{th}}$ centile) rates in second children.

### Methods

Records for 5612 women were drawn from a population-based cohort of routinely collected antenatal healthcare records (2003–2014) linked to measured child body mass index (BMI) age 4–5 years. We applied the parametric G-formula to estimate the effect of hypothetical changes between pregnancy-1 and pregnancy-2 compared to the natural course scenario (without change) on child-2 BMI.

### Results

Observed overweight/obesity and obesity in child-2 at age 4–5 years were 22.2% and 8·5%, respectively. We estimated that if all mothers started pregnancy-2 with BMI 18·5–24·9 kg/m$^2$ and all smokers stopped smoking, then child-2 overweight/obesity and obesity natural course estimates of 22.3% (95% CI 21.2–23.5) and 8·3% (7·6–9·1), would be reduced to 18.5% (17.4–19.9) and 6.2% (5.5–7.0), respectively. For mothers who started pregnancy-1 with BMI 18·5–24·9 kg/m$^2$, if all smokers stopped smoking, child-2 overweight/obesity and obesity natural course estimates of 17.3% (16.0–18.6) and 5·9% (5·0–6·7) would be

Health Research Authority restricts public sharing of the raw data used in this study. To request access conditional on approval from the appropriate institutional ethics, research governance processes and data owners, please email rgoinfo@soton.ac.uk.

**Funding:** This work is supported by an Academy of Medical Sciences and Wellcome Trust Grant [AMS_HOP001\1060 to NAA] and an NIHR Southampton Biomedical Research and University of Southampton Primary Care, Population Sciences and Medical Education PhD studentship to EJT. The funders had no role in study design, data collection and analysis, decision to publish or preparation of the manuscript. KMG is supported by the UK Medical Research Council (MC_UU_12011/4), the National Institute for Health Research (NIHR Senior Investigator (NF-SI-0515-10042), NIHR Southampton 1000DaysPlus Global Nutrition Research Group (17/63/154) and NIHR Southampton Biomedical Research Centre (IS-BRC-1215-20004)), the European Union (Erasmus+ Programme Early Nutrition eAcademy Southeast Asia-573651-EPP-1-2016-1-DE-EPPKA2-CBHE-JP and ImpENSA 598488-EPP-1-2018-1-DE-EPPKA2-CBHE-JP), the British Heart Foundation (RG/15/17/3174), the US National Institute On Aging of the National Institutes of Health (Award No. U24AG047867) and the UK ESRC and BBSRC (Award No. ES/M00919X/1).

**Competing interests:** KMG has received reimbursement for speaking at conferences sponsored by companies selling nutritional products, and is part of an academic consortium that has received research funding from Abbott Nutrition, Nestec, BenevolentAI Bio Ltd. and Danone. The other authors have declared that no competing interests exist.

reduced to 16.0% (14.6–17.3) and 4·9% (4·1–5·7), respectively. For mothers who started pregnancy-1 with BMI $\geq$30 kg/m$^2$, if BMI was 18·5–24·9 kg/m$^2$ prior to pregnancy-2, child-2 overweight/obesity and obesity natural course estimates of 38.6% (34.7–42.3) and 17·7% (15·1–20·9) would be reduced to 31.3% (23.8–40.0) and 12.5 (8.3–17.4), respectively. If BMI was 25.0–29.9 kg/m$^2$ prior to pregnancy-2, these estimates would be 34.5% (29.4–40.4) and 14.6% (11.2–17.8), respectively.

## Conclusion

Interventions supporting women to lose/maintain weight and quit smoking between pregnancies could help reduce rates of overweight/obesity and obesity in second children. The most effective interventions may vary by maternal BMI prior to the first pregnancy.

## Introduction

In 2020, globally, 39 million children under 5 years old had overweight or obesity [1]. In the UK for the 2022/23 school year, the prevalence of obesity in children at age 5–6 years was 9.2% and at age 10–11 years this was 22.7% [2]. The prevalence of obesity was, however, more than double for children living in the most deprived areas compared with those living in the least deprived; for children aged 10–11 years, the prevalences were 30.2% and 13.1% respectively [2].

Children with overweight and obesity suffer from many psychological and physical co-morbidities, including emotional and behavioural disorders [3, 4]. Since overweight and obesity are difficult to reverse once established, the tracking of childhood adiposity into adulthood is of particular concern [5, 6] and once established, overweight and obesity are difficult to reverse [7]. The role of prevention is therefore key and this research considers this further in the context of modifiable exposures after the birth of the first child and before the conception of the second. During this time families are likely to be in frequent contact with healthcare professionals and this could therefore be an important opportunity to focus on the health of the whole family before a further pregnancy [8–10].

The developmental origins of health and disease concept links exposures during the period prior to and immediately following conception and during pregnancy to lasting effects on the developing fetus and subsequent susceptibility to non-communicable disease, including overweight and obesity [11, 12]. Maternal weight and smoking during pregnancy have both been linked to childhood adiposity [13, 14], and changes to maternal weight and smoking behaviour between successive pregnancies have been shown to affect overweight and obesity in the child born in the latter pregnancy [15].

We aimed to quantify the effect of hypothetical scenarios based around maternal weight and smoking change, singly and in combination, between the first and second pregnancy to estimate effects on the prevalence of second child overweight/obesity ($\geq 85^{th}$ centile) and obesity ($\geq 95^{th}$ centile) at age 4–5 years. We then stratified the analyses to examine if scenario results varied by maternal body mass index (BMI) at the start of the first pregnancy, since observed rates of overweight/obesity and obesity in second children differed when stratified in this way (Table 1). In doing so we aimed to ascertain if the most effective scenarios varied depending on a mother's BMI at the start of her first pregnancy.

**Table 1. Observed percentages of obesity and overweight/obesity recorded in second children at the age of 4–5 years, by maternal body mass index (BMI) at the start of the first pregnancy (P1).**

| Maternal BMI at the start of P1 | n | Obesity ($\geq 95^{th}$ centile) at age 4–5 years (%) † | Overweight/obesity ($\geq 85^{th}$ centile) at age 4–5 years (%) † |
|---|---|---|---|
| 18.5–24.9 kg/m² | 3 409 | 5.84 | 17.19 |
| 25–29.9 kg/m² | 1 466 | 10.23 | 25.85 |
| 30 kg/m² or more | 737 | 17.64 | 38.13 |

† In the whole sample, observed child 2 obesity was 8.54%; observed child 2 overweight/obesity was 22.20%

## Methods

### Sample derivation

Data for this analysis were drawn from the SLOPE (Studying Lifecourse Obesity PrEdictors) study [16, 17]. This is a population-based anonymised linked cohort of prospectively collected routine antenatal and birth records for all births registered at the University Hospital Southampton (UHS), Hampshire, UK (accessed 18/06/2018). These records were then linked to child health data from two community National Health Service (NHS) trusts (Solent NHS Trust (accessed 16/08/2018) and Southern Health NHS Foundation Trust (accessed 14/11/2018) who provide child healthcare for the same area, and include National Child Measurement Programme (NCMP) [18] height and weight measurements recorded when children were between 4 and 5 years old.

Records for women with their first two successive singleton live birth pregnancies were included (2003–2014), where a height and weight measurement was recorded for the second child between 4 and 5 years of age. Fig 1 gives the derivation of the sample for this analysis from the maximum available sample (n = 6663). Exclusions were made from the maximum available sample where the first antenatal care appointment (ANA) for either pregnancy took place after 168 days (24 weeks) gestation, as assessed by ultrasound examination performed by a healthcare professional (HCP). These were likely to be high risk pregnancies referred in from elsewhere, and maternal weight measured at the ANA would have included an element of gestational weight gain [19]. Women who had BMI in the underweight range (< 18.5 kg/m²) at the start of one or both of their pregnancies, or who conceived one or both of their pregnancies through assisted reproductive technologies were also excluded to reduce the potential for any unmeasured residual confounding due to changes in health, and because weight gain in women with underweight may be regarded as a positive intervention. Over 84% of the maximum eligible sample of women (n = 5612) were included in this analysis (Fig 1).

This analysis forms part of a research project approved by the University of Southampton Faculty of Medicine Ethics Committee (ID 24433) and the National Health Service Health Research Authority (IRAS 242031). All data were anonymised by the data owners before being accessed by the research team.

### Exposure assessment

Maternal weight (kilograms (kg)) was objectively measured by a midwife at the first ANA for each pregnancy. In the UK, this is recommended to take place by 10 weeks gestation for an uncomplicated pregnancy [20]. Maternal height (metres (m)) and details of maternal smoking

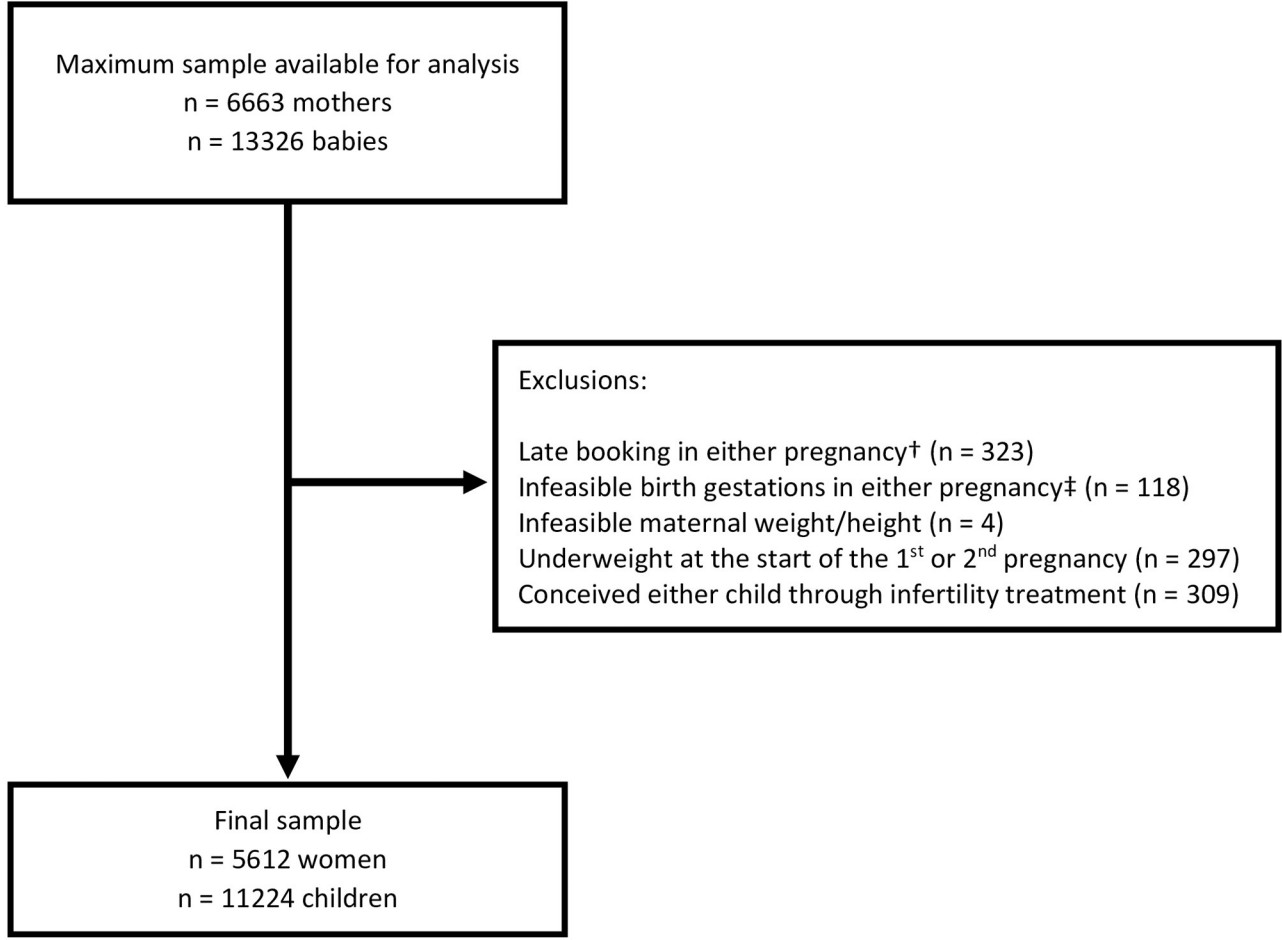

**Fig 1. Flowchart showing the derivation of the sample used in this analysis.** † Over 24 weeks, ‡ Below 22 weeks or over 43 weeks.

status (condensed to smoking/not smoking) were also collected at this appointment by self-report. Maternal BMI was subsequently derived ($kg/m^2$).

BMI was categorised as healthy weight (from 18.5 to 24.9 $kg/m^2$), overweight (from 25 to 29.9 $kg/m^2$) or obese (30 $kg/m^2$ or more). These categories reflect a slightly truncated form of the WHO classifications [21].

### Outcome assessment

Children in state-maintained schools in England have their height and weight measured by school nurses when they are between 4 and 5 years old [18]. Where a measurement was recorded for a second child, BMI was derived ($kg/m^2$) and converted to age- and sex- adjusted BMI $z$-scores using UK 1990 growth reference charts [22]. Overweight/obesity was defined as $\geq 85^{th}$ centile ($z$-score of +1.04) and obesity was defined as $\geq 95^{th}$ centile ($z$-score of +1.65) [23].

### Covariate measurements

Maternal age (in years) was calculated from date of birth prior to extraction of the dataset, to maintain anonymity. Covariates collected by self-report at the first ANA for each pregnancy

included maternal ethnicity (condensed to White, Mixed, Asian, Black/African/Caribbean, Chinese, other and not stated (or not asked)), highest level of maternal education (secondary (GCSE) level or below/ college (A level)/University level or above) and employment status (condensed to employed/not employed). Systolic blood pressure (SBP) (mmHg) was objectively measured by a midwife. Infant birthweight (grams) was objectively measured at birth by a HCP as part of routine care. Details of mode of delivery were also recorded at birth (condensed to caesarean section delivery/not caesarean section delivery). Gestational age was based on a routine dating ultrasound scan undertaken by a HCP which usually takes place between 11 weeks and 2 days and 14 weeks and 1 day gestation [24].

## Hypothetical scenarios

We evaluated scenarios based on change in maternal weight and smoking status between the first and second pregnancy, both individually and in combination, as detailed in Table 2.

Analysis was initially undertaken in the whole sample and was then stratified by maternal BMI at the start of the first pregnancy since, as has already been noted, observed rates of second child overweight/obesity and obesity differed depending on maternal weight at the start of the first pregnancy.

## Statistical analysis

Unadjusted comparisons were carried out using Chi-squared tests for categorical variables and ANOVA for continuous variables.

Table 1 shows observed rates of second child overweight/obesity and obesity by maternal weight at the start of the first pregnancy.

For this analysis the parametric G-formula was applied to estimate the effects of hypothetical scenarios to capture change between the first and second pregnancies (Table 2). The estimated prevalence of overweight/obesity and obesity in second children at age 4 to 5 years was calculated for each scenario using the R package gfoRmula [25].

This method of analysis is based on standard outcome regression modelling and, in brief, involves fitting causally ordered regression models to all the covariates, the exposures and the outcome [26]. It has recently been used to estimate the effects of interventions to prevent stroke [27], myocardial infarction [28] and cognitive impairment in older adults [29].

Models were then constructed to simulate the risk of second child overweight/obesity and obesity under each of the hypothetical scenarios (Table 2), using the observed values of the covariates at baseline and then estimating the values of the other covariates using parametric

**Table 2. The scenarios evaluated for this analysis.**

| Scenario number | Description |
|---|---|
| 0 | Natural course scenario of no change |
| 1 | All women begin their second pregnancy with BMI between 18.5 and 24.9 kg/m$^2$ |
| 2 | All women begin their second pregnancy with BMI between 25 and 29.9kg/m$^2$ |
| 3 | All women begin their second pregnancy with BMI $\geq$ 30 kg/m$^2$ |
| 4 | All smokers stop smoking, and are not active smokers at the first ANA for the second pregnancy |
| 5 | [1] and [4] combined |
| 6 | [2] and [4] combined |
| 7 | [3] and [4] combined |

**Abbreviations:** ANA, antenatal care appointment; BMI, body mass index

models. The values of the covariates were then set to the values specified in the hypothetical scenarios to estimate the predicted risk of overweight/obesity or obesity under each of these hypothetical scenarios. For each model, 250 non-parametric bootstrap samples were generated to estimate 95% confidence intervals.

Maternal ethnicity, highest level of educational attainment and employment status were included as baseline confounders, based on details collected at the start of the first pregnancy.

All analyses were performed using R [30].

## Results

### Characteristics of those included in this study

Table 3 shows characteristics of the women and infants included in this study, by maternal BMI at the start of the first pregnancy. Of women included in this analysis, 60.7% (n = 3409) began their first pregnancy with BMI between 18.5 and 24.9 kg/m$^2$, 26.1% (n = 1466) had BMI between 25 and 29.9 kg/m$^2$, and 13.1% (n = 737) had BMI $\geq$ 30kg/m$^2$ at the start of their first pregnancy.

At the start of their second pregnancy, 79.6% (n = 2714 of 3409) of women maintained a BMI in the range of 18.5–24.9 kg/m$^2$, 56.1% (n = 823 of 1466) maintained a BMI in the range 25–29.9 kg/m$^2$, and 87.2% (n = 643 of 737) maintained a BMI $\geq$ 30kg/m$^2$.

Of those who started their first pregnancy with BMI $\geq$ 30 kg/m$^2$, 1.5% (n = 11) started their second pregnancy with a BMI in the range of 18.5–24.9 kg/m$^2$, and 11.3% (n = 83) with a BMI in the range of 25–29.9 kg/m$^2$. Overall, 60.7% (n = 447) of women with BMI $\geq$ 30 kg/m$^2$ gained weight from the start of their first to the start of their second pregnancy; 27.0% (n = 199) exhibited significant weight gain of $\geq$ 3 kg/m$^2$. Whilst 36.2% (n = 267) of women in this category lost weight from the start of their first to the start of their second pregnancy, only 12.9% (n = 95) lost $\geq$ 3 kg/m$^2$.

Second born children with obesity or overweight/obesity at age 4–5 years were more likely to have mothers who started their first or second pregnancies with obesity or overweight/obesity, who exhibited high weight gain ($\geq$ 3 kg/m$^2$) between pregnancies and who were smoking at the start of their first or second pregnancy.

Table 4 presents the analyses for each hypothetical scenario (Table 2), in the whole sample and then stratified by maternal BMI range at the start of the first pregnancy.

### Results for the whole sample

In the whole sample, scenarios 1 and 5 showed the greatest impact in reducing the prevalence of obesity and overweight/obesity in second children at age 4 to 5 years. Scenarios 1 and 5 both include starting the second pregnancy with BMI 18.5–24.9 kg/m$^2$; when considered as the only change (scenario 1), there would be a 20.1% (95% confidence interval (CI) 16.0, 24.5) population risk reduction in obesity to 6.6% (5.9, 7.4) and a 13.7% (10.9, 16.3) population risk reduction in overweight/obesity to 19.2% (18.2, 20.7), compared to the NCE (scenario 0).

### Results for women who started their first pregnancy with BMI 18.5–24.9 kg/m$^2$

In the sample of women who started their first pregnancy with BMI 18.5–24.9 kg/m$^2$, scenarios 4 and 5 had the greatest impact in reducing the prevalence of obesity and overweight/obesity in second children at age 4 to 5 years. Scenarios 4 and 5 both include smokers having stopped smoking by the first ANA for the second pregnancy; when considered as the only change there would be a 16.7% (10.0, 23.2) population risk reduction in obesity to 4.9% (4.1, 5.7) and a 7.4%

**Table 3. Maternal, birth and child characteristics by maternal body mass index (BMI) at the start of the first pregnancy.** All figures are proportions (%), unless otherwise stated.

| | Maternal BMI at the start of the first pregnancy | | | |
| --- | --- | --- | --- | --- |
| | 18.5 to 24.9kg/m² n = 3 409 | 25 to 29.9kg/m² n = 1 466 | ≥ 30kg/m² n = 737 | *p*-value[1] |
| Collected at the start of the first pregnancy | | | | |
| Age, years (mean, SD) | 25.9 (5.5) | 26.1 (5.2) | 26.1 (5.2) | 0.199 |
| BMI, kg/m² (mean, SD) | 22.0 (1.7) | 27.0 (1.4) | 34.3 (4.0) | < 0.001 |
| SBP mmHg (mean, SD) | 108.7 (10.6) | 112.4 (10.5) | 117.7 (10.8) | < 0.001 |
| (missing records) | (n = 39) | (n = 12) | (n = 3) | |
| Smoking status: | | | | 0.298 |
| Not smoking | 85.2 | 85.3 | 83.0 | |
| Smoking | 14.8 | 14.7 | 17.0 | |
| Collected at the start of the second pregnancy | | | | |
| Age, years (mean, SD) | 28.8 (5.5) | 29.0 (5.2) | 29.0 (5.2) | 0.235 |
| BMI, kg/m² (mean, SD) | 23.1 (2.7) | 28.4 (3.5) | 35.2 (5.3) | < 0.001 |
| SBP mmHg (mean, SD) | 107.7 (10.2) | 112.0 (10.4) | 117.0 (10.7) | < 0.001 |
| (missing records) | (n = 59) | (n = 34) | (n = 10) | |
| Smoking status: | | | | 0.085 |
| Not smoking | 86.8 | 87.3 | 84.0 | |
| Smoking | 13.2 | 12.8 | 16.0 | |
| Recorded at the first birth | | | | |
| Delivery by Caesarean section (%) | 17.9 | 24.6 | 31.9 | < 0.001 |
| (missing records) | (n = 9) | (n = 2) | (n = 1) | |
| Birthweight, grams (mean, SD) | 3342 (510) | 3411 (532) | 3500 (562) | < 0.001 |
| Gestational age, days (mean, SD)[3] | 280 (12) | 280 (13) | 281 (14) | 0.049 |
| Recorded at the second birth | | | | |
| Delivery by Caesarean section (%) | 14.7 | 21.5 | 28.0 | < 0.001 |
| (missing records) | (n = 54) | (n = 21) | (n = 5) | |
| Birthweight, grams (mean, SD) | 3497 (501) | 3587 (502) | 3681 (551) | < 0.001 |
| Gestational age, days (mean, SD)[3] | 280 (10) | 281 (10) | 281 (11) | 0.003 |
| Other details | | | | |
| Ethnicity: | | | | < 0.001 |
| White | 87.8 | 90.3 | 95.0 | |
| Mixed | 0.9 | 0.9 | 1.1 | |
| Asian | 4.6 | 2.9 | 0.9 | |
| Black/African/Caribbean | 1.2 | 1.7 | 0.7 | |
| Chinese | 0.6 | 0.1 | 0.0 | |
| Other | 0.7 | 0.4 | 0.3 | |
| Not specified | 4.2 | 3.8 | 2.0 | |
| Highest education level at baseline: | | | | < 0.001 |
| University or above | 30.2 | 25.2 | 21.0 | |
| College (A levels) | 36.6 | 40.5 | 47.2 | |
| Secondary or below (GCSEs) | 33.2 | 34.2 | 31.8 | |
| Employment at baseline: | | | | 0.022 |
| In employment | 83.5 | 85.5 | 87.1 | |
| Not in employment | 16.5 | 14.5 | 12.9 | |
| (missing records) | (n = 11) | (n = 5) | (n = 1) | |

(*Continued*)

**Table 3.** (Continued)

| | Maternal BMI at the start of the first pregnancy | | | |
| --- | --- | --- | --- | --- |
| | 18.5 to 24.9kg/m$^2$ n = 3 409 | 25 to 29.9kg/m$^2$ n = 1 466 | ≥ 30kg/m$^2$ n = 737 | *p*-value[1] |
| Weight change between pregnancies: | | | | < 0.001 |
| Weight stable (Gain or loss up to 1kg/m$^2$) | 41.3 | 28.8 | 23.7 | |
| Weight loss (Loss ≥ 1kg/m$^2$) | 12.7 | 20.1 | 26.3 | |
| Moderate gain (Gain ≥ 1 and < 3kg/m$^2$) | 30.1 | 25.9 | 22.9 | |
| High gain (Gain ≥ 3kg/m$^2$) | 15.9 | 25.3 | 27.0 | |
| Length of the interpregnancy interval[2] | | | | 0.174 |
| < 12 months | 17.5 | 14.9 | 19.0 | |
| 12 to < 24 months | 38.3 | 38.6 | 36.9 | |
| 24 to < 36 months | 23.3 | 25.8 | 24.2 | |
| 36 months or more | 20.9 | 20.7 | 19.9 | |
| First child at age 4–5 years | | | | |
| With obesity (≥ 95$^{th}$ centile) | 5.7 | 8.3 | 12.7 | < 0.001 |
| Without obesity | 94.3 | 91.7 | 87.3 | |
| (missing records) | (n = 1451) | (n = 669) | (n = 313) | |
| With overweight/obesity (≥ 85$^{th}$ centile) | 15.9 | 23.8 | 31.1 | < 0.001 |
| Without overweight obesity | 84.1 | 76.2 | 68.9 | |
| (missing records) | (n = 1451) | (n = 669) | (n = 313) | |
| Second child at age 4–5 years | | | | |
| With obesity (≥ 95$^{th}$ centile) | 5.8 | 10.2 | 17.6 | < 0.001 |
| Without obesity | 94.2 | 89.8 | 82.4 | |
| With overweight/obesity (≥ 85$^{th}$ centile) | 17.2 | 25.9 | 38.1 | < 0.001 |
| Without overweight obesity | 82.8 | 74.1 | 61.9 | |

**Abbreviations:** BMI, body mass index; A level, Advanced level; GCSE, General Certificate of Secondary Education; SBP, systolic blood pressure; SD, standard deviation

[1]. *p*-values calculated using ANOVA for continuous and Chi-squared tests for categorical variables

[2]. From the birth of the first child to the conception of the second

[3]. 280 days gestation is equivalent to 40 weeks

(3.9, 11.0) population risk reduction in overweight/obesity to 16.0% (14.6, 17.3), compared to the NCE (scenario 0).

### Results for women who started their first pregnancy with BMI 25–29.9 kg/m$^2$

In the sample of women who started their first pregnancy with BMI 25–29.9 kg/m$^2$, none of the scenarios produced significant results, compared to the NCE (scenario 0).

### Results for women who started their first pregnancy with BMI ≥ 30 kg/m$^2$

In the sample of women who started their first pregnancy with BMI ≥ 30 kg/m$^2$ the scenarios with the greatest impact in reducing the prevalence of obesity and overweight/obesity in second children at age 4 to 5 years all included weight reduction by the start of the second pregnancy. When starting the second pregnancy with a BMI of 18.5–24.9 kg/m$^2$ as the only change there would be a 29.4% (4.7, 53.4) population risk reduction in obesity to 12.5% (8.3, 17.4) and a 18.9% (0.8, 36.9) population risk reduction in overweight/obesity to 31.3% (23.8, 39.9), compared to the NCE (scenario 0). When starting the second pregnancy with a BMI of 25–29.9 kg/

**Table 4. The percentages of second children with obesity and overweight/obesity at age 4–5 years under different hypothetical scenarios, together with population risk ratios and population mean differences.**

| Scenario Number and Description | 2nd children at age 4–5 years % (95% CI) | | Population risk ratio (95% CI) | | Population mean difference (95% CI) | |
|---|---|---|---|---|---|---|
| | Obesity[†] | Overweight/ obesity[†] | Obesity[†] | Overweight/ obesity[†] | Obesity[†] | Overweight/ obesity[†] |
| Analysis in the whole sample | | | | | | |
| [0] Natural course[1] | 8.30 (7.59, 9.05) | 22.22 (21.21, 23.47) | 1.00 | 1.00 | 0.00 | 0.00 |
| [1] BMI between 18.5 and 24.9 kg/m² at the start of P2 | 6.63 (5.94, 7.42) | 19.19 (18.16, 20.66) | 0.80 (0.75, 0.84) | 0.86 (0.84, 0.89) | -1.67 (-2.08, -1.33) | -3.04 (-3.67, -2.38) |
| [2] BMI between 25 and 29.9 kg/m² at the start of P2 | 8.32 (7.63, 9.11) | 22.65 (21.65, 23.90) | 1.00 (0.99, 1.01) | 1.02 (1.01, 1.03) | 0.02 (-0.07, 0.12) | 0.43 (0.30, 0.58) |
| [3] BMI ≥ 30 kg/m² at the start of P2 | 10.53 (9.50, 11.60) | 26.85 (25.29, 28.44) | 1.27 (1.21, 1.34) | 1.21 (1.17, 1.25) | 2.23 (1.71, 2.81) | 4.63 (3.70, 5.62) |
| [4] All smokers quit[‡] | 7.72 (6.96. 8.56) | 21.43 (20.41, 22.73) | 0.93 (0.89, 0.98) | 0.96 (0.94, 0.99) | -0.58 (-0.92, -0.18) | -0.80 (-1.36, 0.32) |
| [5] Scenarios [1] and [4] combined | 6.16 (5.50, 6.95) | 18.46 (17.42, 19.86) | 0.74 (0.69, 0.79) | 0.83 (0.80, 0.87) | -2.14 (-2.60, -1.68) | -3.76 (-4.49, -3.00) |
| [6] Scenarios [2] and [4] combined | 7.74 (7.00, 8.60) | 21.84 (20.83, 23.23) | 0.93 (0.89, 0.98) | 0.98 (0.96, 1.01) | -0.56 (-0.89, -0.13) | -0.39 (-0.91, 0.12) |
| [7] Scenarios [3] and [4] combined | 9.82 (8.73, 10.96) | 25.96 (24.33, 27.72) | 1.18 (1.10, 1.28) | 1.17 (1.11, 1.22) | 1.52 (0.82, 2.33) | 3.73 (2.51, 4.84) |
| Analysis in the group of women who started their first pregnancy with BMI in the healthy range (between 18.5 and 24.9 kg/m²) | | | | | | |
| [0] Natural course[2] | 5.85 (5.03, 6.68) | 17.32 (15.96, 18.58) | 1.00 | 1.00 | 0.00 | 0.00 |
| [1] BMI between 18.5 and 24.9 kg/m² at the start of P2 | 5.52 (4.67, 6.34) | 16.86 (15.49, 18.20) | 0.94 (0.91, 0.97) | 0.97 (0.96, 0.99) | -0.32 (-0.49, -0.17) | -0.46 (-0.66, -0.24) |
| [2] BMI between 25 and 29.9 kg/m² at the start of P2 | 7.15 (6.14, 8.28) | 19.56 (17.96, 21.22) | 1.22 (1.12, 1.32) | 1.13 (1.07, 1.19) | 1.30 (0.73, 1.88) | 2.24 (1.23, 3.21) |
| [3] BMI ≥ 30 kg/m² at the start of P2 | 11.35 (8.41, 15.12) | 25.25 (21.35, 29.41) | 1.94 (1.44, 2.57) | 1.46 (1.24, 1.68) | 5.50 (2.68, 8.94) | 7.93 (4.12, 11.83) |
| [4] All smokers quit[‡] | 4.87 (4.07, 5.70) | 16.03 (14.60, 17.25) | 0.83 (0.77, 0.90) | 0.93 (0.89, 0.96) | -0.97 (-1.38, -0.56) | -1.29 (-1.91, -0.64) |
| [5] Scenarios [1] and [4] combined | 4.60 (3.85, 5.43) | 15.60 (14.19, 16.80) | 0.79 (0.72, 0.85) | 0.90 (0.86, 0.93) | -1.25 (-1.64, -0.87) | -1.72 (-2.42, -1.10) |
| [6] Scenarios [2] and [4] combined | 5.99 (4.99, 7.17) | 18.17 (16.50, 19.77) | 1.03 (0.91, 1.16) | 1.05 (0.98, 1.12) | 0.15 (-0.55, 0.89) | 0.85 (-0.27, 2.09) |
| [7] Scenarios [3] and [4] combined | 9.65 (7.01, 13.29) | 23.62 (19.78, 27.66) | 1.65 (1.21, 2.24) | 1.36 (1.15, 1.58) | 3.81 (1.30, 7.36) | 6.30 (2.63, 10.12) |
| Analysis in the group of women who started their first pregnancy with BMI in the overweight range (between 25 and 29.9 kg/m²) | | | | | | |
| [0] Natural course[3] | 9.81 (8.38, 11.27) | 25.65 (23.50, 28.15) | 1.00 | 1.00 | 0.00 | 0.00 |
| [1] BMI between 18.5 and 24.9 kg/m² at the start of P2 | 9.80 (7.48, 12.20) | 25.27 (21.91, 29.25) | 1.00 (0.83, 1.18) | 0.99 (0.89, 1.10) | -0.01 (-1.50, 1.71) | -0.38 (-3.02, 2.60) |
| [2] BMI between 25 and 29.9 kg/m² at the start of P2 | 9.81 (8.29, 11.29) | 25.59 (23.38, 28.07) | 1.00 (0.97, 1.02) | 1.00 (0.98, 1.01) | 0.00 (-0.30, 0.21) | -0.05 (-0.46, 0.31) |
| [3] BMI ≥ 30 kg/m² at the start of P2 | 9.82 (8.11, 11.41) | 25.86 (22.79, 28.44) | 1.00 (0.91, 1.09) | 1.01 (0.94, 1.07) | 0.00 (-0.90, 0.80) | 0.21 (-1.50, 1.75) |
| [4] All smokers quit[‡] | 9.62 (8.04, 11.39) | 25.28 (22.94, 28.03) | 0.98 (0.91, 1.05) | 0.99 (0.95, 1.02) | -0.19 (-0.89, 0.46) | -0.37 (-1.24, 0.56) |
| [5] Scenarios [1] and [4] combined | 9.62 (7.17, 12.31) | 24.91 (21.47, 28.70) | 0.98 (0.78, 1.18) | 0.97 (0.86, 1.08) | -0.20 (-2.02, 1.75) | -0.74 (-3.64, 2.05) |
| [6] Scenarios [2] and [4] combined | 9.62 (8.02, 11.39) | 25.22 (22.82, 27.87) | 0.98 (0.90, 1.05) | 0.98 (0.94, 1.02) | -0.19 (-0.97, 0.51) | -0.42 (-1.43, 0.56) |
| [7] Scenarios [3] and [4] combined | 9.63 (7.82, 11.15) | 25.49 (22.36, 28.51) | 0.98 (0.85, 1.09) | 0.99 (0.91, 1.06) | -0.18 (-1.41, 0.84) | -0.16 (-2.33, 1.50) |
| Analysis in the group of women who started their first pregnancy with BMI in the obese range (≥ 30 kg/m²) | | | | | | |

*(Continued)*

**Table 4.** (Continued)

| Scenario Number and Description | 2nd children at age 4–5 years % (95% CI) | | Population risk ratio (95% CI) | | Population mean difference (95% CI) | |
|---|---|---|---|---|---|---|
| | Obesity[†] | Overweight/ obesity[†] | Obesity[†] | Overweight/ obesity[†] | Obesity[†] | Overweight/ obesity[†] |
| [0] Natural course[4] | 17.74 (15.05, 20.89) | 38.60 (34.74, 42.28) | 1.00 | 1.00 | 0.00 | 0.00 |
| [1] BMI between 18.5 and 24.9 kg/m² at the start of P2 | 12.52 (8.28, 17.41) | 31.31 (23.76, 39.92) | 0.71 (0.47, 0.95) | 0.81 (0.63, 0.99) | -5.22 (-9.65, -0.88) | -7.29 (-14.70, -0.30) |
| [2] BMI between 25 and 29.9 kg/m² at the start of P2 | 14.62 (11.17, 17.81) | 34.45 (29.39, 40.42) | 0.82 (0.65, 0.97) | 0.89 (0.78, 1.00) | -3.12 (-6.20, -0.50) | -4.15 (-8.70, -0.17) |
| [3] BMI ≥ 30 kg/m² at the start of P2 | 17.94 (15.19, 21.17) | 38.90 (34.87, 42.73) | 1.01 (1.00, 1.02) | 1.00 (1.00, 1.01) | 0.21 (0.04, 0.36) | 0.30 (0.01, 0.55) |
| [4] All smokers quit[‡] | 18.06 (15.30, 21.45) | 39.27 (35.08, 43.39) | 1.02 (0.94, 1.10) | 1.02 (0.97, 1.07) | 0.32 (-0.97, 1.70) | 0.67 (-1.07, 2.72) |
| [5] Scenarios [1] and [4] combined | 12.76 (8.49, 17.33) | 31.92 (23.89, 40.80) | 0.72 (0.46, 0.97) | 0.83 (0.64, 1.01) | -4.98 (-9.53, -0.53) | -6.68 (-14.30, 0.49) |
| [6] Scenarios [2] and [4] combined | 14.89 (11.37, 18.26) | 35.09 (29.60, 41.15) | 0.84 (0.65, 1.00) | 0.91 (0.78, 1.02) | -2.85 (-6.08, -0.04) | -3.51 (-8.52, 0.67) |
| [7] Scenarios [3] and [4] combined | 18.27 (15.46, 21.72) | 39.57 (35.10, 43.71) | 1.03 (0.95, 1.11) | 1.03 (0.98, 1.07) | 0.53 (-0.88, 2.00) | 0.97 (-0.72, 2.97) |

**Abbreviations:** BMI, body mass index; CI, confidence intervals; P2, the second pregnancy

[†] Obesity defined as ≥ 95th centile; Overweight/obesity defined as ≥ 85th centile

[‡] No active smokers at the first antenatal care appointment for the second pregnancy

1. Observed obesity 8.54%; observed overweight/obesity 22.20%

2. Observed obesity 5.84%; observed overweight/obesity 17.19%

3. Observed obesity 10.23%; observed overweight/obesity was 25.85%

4. Observed obesity 17.64%; observed overweight/obesity was 38.13%

m² as the only change there would be a 17.6% (2.6, 35.1) population risk reduction in obesity to 14.6% (11.2, 17.8) and a 10.8% (0.4, 22.0) population risk reduction in overweight/obesity to 34.5% (29.4, 40.4), compared to the NCE (scenario 0).

## Discussion

According to this analysis, changes after the birth of the first child and before the conception of the second could play an important role in reducing overweight/obesity and obesity rates in second children at the start of primary school. The most effective scenarios may vary by maternal BMI at the start of the first pregnancy. Weight maintenance in women who start their first pregnancy with BMI in the range of 18.5–24.9 kg/m² is important, but for this group of women smoking cessation was the scenario with the greatest effect. Conversely, whilst smoking cessation would be beneficial for many reasons in the group of women who started their first pregnancy with BMI in the obese range, weight loss in the interconception period had the greatest impact.

Cessation of smoking prior to pregnancy is known to be beneficial since maternal smoking increases the risk of miscarriage, stillbirth, prematurity and fetal growth restriction [31]. Longer term, where a fetus has been exposed to maternal smoking, there is a higher risk of childhood overweight and obesity [14], and children of maternal smokers have higher central adiposity than the children of non-smokers [32]. In the UK, HCPs are able to refer pregnant smokers to smoking cessation services and this should certainly be encouraged. Intervention

during the interpregnancy interval may help prevent re-uptake or inception of smoking prior to the next pregnancy. Intervention at the earliest stage may also help to reduce the risk of other adverse outcomes, for example SGA birth, which in turn has been associated with offspring adiposity [6].

Childhood obesity risk is increased in mothers with obesity, and animal models have demonstrated that maternal over-nutrition from a high fat diet can predict obesity in adulthood, independent of post-natal diet [33, 34]. The worldwide increasing prevalence of obesity in women of reproductive age is a major concern, and may impact not only maternal health but also that of any future children [35]. Overweight and obesity in pregnancy increases the risk of pregnancy and birth related complications [36], which can, in turn, lead to increased risks of childhood overweight and obesity. These complications include developing gestational diabetes [35, 37], having an infant with high birthweight or who is born large for their gestational age [33, 38] and requiring a caesarean section delivery [36, 39].

Only 35.7% of the women in this analysis (n = 2,005) remained weight stable between pregnancies (defined here as a difference of a gain or loss of up to 1 kg/m$^2$). More than a quarter of mothers who started their first pregnancy with BMI in either the overweight or obese range showed high weight gain between their first two pregnancies (defined here as a gain of 3 kg/m$^2$ or more). These categories have been selected to be consistent with previous literature as a way of enabling a straightforward comparison of weight change between pregnancies [40]. This analysis provides evidence to suggest that more awareness of the benefits of returning to a healthy weight after pregnancy could have an effect on the long-term health of subsequent children, particularly when a mother started her first pregnancy with obesity. It is, of course, not as simple as promoting awareness. There are other barriers to returning to a healthy weight [41], shaped by wider socio-economic and environmental factors. This includes the impact of the built environment, a potential lack of available greenspace, the cost of healthy foods, access to unhealthy food outlets, the necessity of travelling further distances to retail outlets selling healthier options and a lack of time and resources to acquire, cook or eat healthier food items. Strategies which support women returning to, or reaching, a healthy weight need to take account of the fact that high weight loss between pregnancies has itself been associated with childhood adiposity [15] and also with increased risk of other adverse outcomes including preterm and SGA birth [40, 42, 43]. SGA birth in conjunction with rapid postnatal catch up growth, is also associated with childhood adiposity [33].

We found that in women who started their first pregnancies with BMI of 18.5–24.9 kg/m$^2$, smoking cessation combined with weight maintenance was the most effective scenario in reducing the risk of offspring overweight or obesity. Whilst smoking cessation is also to be strongly encouraged in women who start their first pregnancies with BMI in the obese range, here reducing weight between pregnancies was the most effective scenario to prevent second child overweight/obesity and obesity. Interventions based around weight maintenance and a steady return to pre-pregnancy weight prior to conceiving again are therefore to be encouraged.

Optimising maternal health between consecutive pregnancies will not only ensure that a mother is well prepared for her next pregnancy and birth but may also help to improve the longer-term health of her child [44]. Interventions during the post-partum period, which is the preconception period for a subsequent pregnancy, may be possible since a mother is still in frequent contact with HCPs during that time. It may be also be possible to incorporate timely interventions within Women's Health Hubs in England, which aim to provide easily accessible and holistic care to women throughout their lives [45].

A recent systematic review which considered the effectiveness of interventions delivered by HCPs during the period from conception to a child's second birthday (the first 1,000 days) did

find evidence that a few were effective in reducing childhood adiposity [46]. Whether HCPs have the time during existing appointments with young families in which to deliver such interventions will also need to be considered, and any interventions which are based around maternal weight will need careful consideration to ensure that existing inequalities based around access to healthy food, for example, are not widened [47, 48].

### Strengths and limitations of this study

Strengths of this study include drawing on data from the SLOPE study, a large population-based cohort that is representative of the local population. It includes women from all socio-economic and ethnic backgrounds. The analysis adjusted for several key confounders, a number of which were objectively measured by HCPs. Maternal BMI measurements, which form part of the exposures considered, were based on objectively measured maternal weight at the start of each pregnancy. Outcome measurements were also objectively measured and recorded by school nurses as part of the NCMP in schools.

There are some limitations, including the collection of information by self-report. This includes maternal smoking status at the start of each pregnancy and raises the possibility of non-disclosure and information bias. In addition, the NCMP is an opt-out scheme and some parents may not have allowed their children to be part of the process. Participation rates are high, however, with over 95% of children participating annually [49].

Whilst the application of the parametric G-formula in this analysis enabled the consideration of a number of hypothetical scenarios capturing change between pregnancies and enabled a comparison of scenarios based on maternal BMI at the start of the first pregnancy, future analyses should consider additional scenarios.

### Conclusion

Interventions after the birth of the first child and before the conception of the second child could reduce the prevalence of overweight/obesity and obesity in second and subsequent children at the age of 4 to 5 years. The most effective interventions may vary by maternal BMI at the start of the first pregnancy. The time between pregnancies, when women are still in regular contact with HCPs is an important opportunity for well-timed interventions which could impact the health of the whole family. Using maternal BMI at the start of the first pregnancy to group women could enable HCPs to identify groups where additional support might be offered. Support to encourage mothers who had healthy weight at the start of their first pregnancy to quit smoking and maintain their weight, and support mothers who had obesity at the start of their first pregnancy to lose weight between pregnancies could have the greatest impact the health of mothers and their children.

### Acknowledgments

We thank David Cable (Electronic Patient Records Implementation and Service Manager) and Florina Borca (Senior Information Analyst R&D) at University Hospital Southampton for support in accessing the data used in this study. We thank Gareth Edwards (Business Intelligence Developer at Solent NHS Trust) as well as the Research and Development and data team at Southern Health NHS Foundation Trust for their help in access the early life data used in this study.

### Author Contributions

**Conceptualization:** Nisreen A. Alwan.

**Data curation:** Nida Ziauddeen.

**Formal analysis:** Elizabeth J. Taylor, Ann Berrington, Keith M. Godfrey, Nisreen A. Alwan.

**Funding acquisition:** Nisreen A. Alwan.

**Methodology:** Elizabeth J. Taylor, Nisreen A. Alwan.

**Project administration:** Nisreen A. Alwan.

**Supervision:** Ann Berrington, Keith M. Godfrey, Nisreen A. Alwan.

**Writing – original draft:** Elizabeth J. Taylor.

**Writing – review & editing:** Elizabeth J. Taylor, Nida Ziauddeen, Ann Berrington, Keith M. Godfrey, Nisreen A. Alwan.

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
