## [Decision Letter · Decision Letter 0]

31 Jul 2024

PONE-D-24-08911Quantifying the effect of interpregnancy maternal weight and smoking status changes on childhood overweight and obesity in a UK population-based cohortPLOS ONE

Dear Dr. Ziauddeen,

Thank you for submitting your manuscript to PLOS ONE. After careful consideration, we feel that it has merit but does not fully meet PLOS ONE’s publication criteria as it currently stands. Therefore, we invite you to submit a revised version of the manuscript that addresses the points raised during the review process.

We look forward to receiving your revised manuscript.

Kind regards,

Engelbert A. Nonterah, MD, PhD

Academic Editor

PLOS ONE

2. We noted in your submission details that a portion of your manuscript may have been presented or published elsewhere. [This work was presented at the virtual Lancet Public Health Science conference in November 2021 and the submitted abstract was published (https://doi.org/10.1016/S0140-6736(21)02627-1). This work originally formed part of the thesis for a Doctor of Philosophy degree undertaken by Elizabeth Taylor and funded by NIHR Southampton Biomedical Research Centre and University of Southampton Primary Care, Population Sciences and Medical Education PhD studentship. The thesis is under embargo until October 2024.] Please clarify whether this [conference proceeding or publication] was peer-reviewed and formally published. If this work was previously peer-reviewed and published, in the cover letter please provide the reason that this work does not constitute dual publication and should be included in the current manuscript.

Please update your Data Availability statement in the submission form accordingly

Reviewers' comments:

Reviewer's Responses to Questions

**Comments to the Author**

1. Is the manuscript technically sound, and do the data support the conclusions?

Reviewer #1: Yes

Reviewer #2: Yes

2. Has the statistical analysis been performed appropriately and rigorously? 

Reviewer #1: Yes

Reviewer #2: Yes

3. Have the authors made all data underlying the findings in their manuscript fully available?

Reviewer #1: Yes

Reviewer #2: No

4. Is the manuscript presented in an intelligible fashion and written in standard English?

Reviewer #1: Yes

Reviewer #2: Yes

5. Review Comments to the Author

Reviewer #1: Reviewer’s comments

Introduction

The introduction is clear and concise, but the literature cited is very global, none on the context where the study is conducted. Suggestion: include more contextual information including the problem and the gaps that the study addresses.

Methods

Very well and clearly detailed.

Results

Well detailed, aligned with the objectives and statistical analysis. However, quite difficult to follow, the author should add sub-topics representing the different sub-objectives / scenarios for clarity.

Reviewer #2: An excellent paper. Well written, conceived and interesting. Results support the need for good pre- / inter-conception strategies to optimise maternal health and mitigate childhood risks.

A few minor comments and suggests for the author(s):

Methods:

1. What is your rationale for excluding women who conceived via artificial reproductive therapy? This should be more clearly described and discussed in the paper. With increasing maternal age, more women are seeking to conceive via ART. Often BMI restrictions in the private sector are less closely adhered to than in the NHS. It is important that full representation of the pregnant population is described.

2. Exposure assessment - please state that the chosen BMI classes reflect a truncated WHO classification (REF).

Results:

1. Table 3: gestational age could be more meaningful to present as weeks and days depending on readership. It may be clearer to include both this in additional to gestational age in days.

2. Table 3: please check that units are clearly presented for all variables (e.g) weight change between pregnancies (%) in table 3.

3. Abbreviations 3 and 4 are not described in the footnotes for table 3, please amend.

4. What was your rationale for significant weight gain (3 kg or more)? Please describe why in the manuscript.

Discussion:

Well thought through and presented. Further discussion could include influence of recent maternity and public health strategies to support your proposed interventions. For example, the Women’s Health Strategy for England.

https://www.gov.uk/government/publications/womens-health-hubs-information-and-guidance/womens-health-hubs-core-specification

6. PLOS authors have the option to publish the peer review history of their article (what does this mean?). If published, this will include your full peer review and any attached files.

Reviewer #1: No

Reviewer #2: **Yes: **Dr Kelly-Ann Eastwood

---

## [Author Response · Author response to Decision Letter 0]

20 Sep 2024

Thank you, we have updated the formatting to meet PLOS ONE’s requirements. 

2. We noted in your submission details that a portion of your manuscript may have been presented or published elsewhere. [This work was presented at the virtual Lancet Public Health Science conference in November 2021 and the submitted abstract was published (https://doi.org/10.1016/S0140-6736(21)02627-1). This work originally formed part of the thesis for a Doctor of Philosophy degree undertaken by Elizabeth Taylor and funded by NIHR Southampton Biomedical Research Centre and University of Southampton Primary Care, Population Sciences and Medical Education PhD studentship. The thesis is under embargo until October 2024.] Please clarify whether this [conference proceeding or publication] was peer-reviewed and formally published. If this work was previously peer-reviewed and published, in the cover letter please provide the reason that this work does not constitute dual publication and should be included in the current manuscript.

The abstract was reviewed by the conference organising committee for decision making on acceptance of abstract to the conference programme. The short abstract was published in a supplement of the Lancet which is the usual practice for this conference’s accepted abstracts every year. This does not constitute dual publication as the conference abstract does not include the full introduction, methods, results and discussion included in the paper. 

Please update your Data Availability statement in the submission form accordingly

We have edited the data availability statement as follows: The study’s ethical approval from the Faculty of Medicine Ethics Committee, University of Southampton and the Health Research Authority restricts public sharing of the raw data used in this study. To request access conditional on approval from the appropriate institutional ethics, research governance processes and data owners, please email rgoinfo@soton.ac.uk.

Thank you, we have reviewed the reference list as suggested.

Reviewers' comments:

Reviewer #1: 

Introduction: The introduction is clear and concise, but the literature cited is very global, none on the context where the study is conducted. Suggestion: include more contextual information including the problem and the gaps that the study addresses.

Thank you. Further detail has now been included in the introduction to address the reviewer’s comments (lines 50-54 and 58-63). These pick up on the specific prevalence of children with obesity in the UK and add more contextual detail.

Methods: Very well and clearly detailed. Thank you.

Results: Well detailed, aligned with the objectives and statistical analysis. However, quite difficult to follow, the author should add sub-topics representing the different sub-objectives / scenarios for clarity. 

Thank you, sub-headings have now been added to aid clarity.

Reviewer #2: 

An excellent paper. Well written, conceived and interesting. Results support the need for good pre- / inter-conception strategies to optimise maternal health and mitigate childhood risks. Thank you.

A few minor comments and suggests for the author(s):

Methods:

1. What is your rationale for excluding women who conceived via artificial reproductive therapy? This should be more clearly described and discussed in the paper. With increasing maternal age, more women are seeking to conceive via ART. Often BMI restrictions in the private sector are less closely adhered to than in the NHS. It is important that full representation of the pregnant population is described.

Thank you. Women who conceived one or both of their first two pregnancies using assisted reproductive technologies have been excluded to reduce any potential for residual confounding because of unmeasured health changes associated with this very heterogeneous group of interventions. Few exclusions were made for this reason (n = 309). The manuscript has been updated to be more specific for the reason for this exclusion (lines 100-102).

2. Exposure assessment - please state that the chosen BMI classes reflect a truncated WHO classification (REF).

Thank you, this has now been included as suggested (lines 117-118).

Results:

1. Table 3: gestational age could be more meaningful to present as weeks and days depending on readership. It may be clearer to include both this in additional to gestational age in days.

Thank you. For clarity, a footnote has been added to the Table to state that 280 days is equivalent to 40 weeks gestation.

2. Table 3: please check that units are clearly presented for all variables (e.g) weight change between pregnancies (%) in table 3.

Thank you, the second line of the table heading was omitted in error and has now been re-instated to show “All figures are proportions (%) unless stated otherwise.”

3. Abbreviations 3 and 4 are not described in the footnotes for table 3, please amend.

Thank you, these have now been replaced with a description, to be consistent with the entries for the second child. The abbreviations in the table have also been updated.

4. What was your rationale for significant weight gain (3 kg or more)? Please describe why in the manuscript.

Thank you. A brief description of the rationale with appropriate references has now been added to the manuscript (lines 291-293).

Discussion:

Well thought through and presented. Further discussion could include influence of recent maternity and public health strategies to support your proposed interventions. For example, the Women’s Health Strategy for England.

https://www.gov.uk/government/publications/womens-health-hubs-information-and-guidance/womens-health-hubs-core-specification

Thank you very much. A brief reference to this has now been included in the paper in the context of delivering interventions (lines 317-319).

---

## [Editor Report · Decision Letter 1]

24 Sep 2024

Quantifying the effect of interpregnancy maternal weight and smoking status changes on childhood overweight and obesity in a UK population-based cohort

PONE-D-24-08911R1

Dear Dr. Nida Ziauddeen,

We’re pleased to inform you that your manuscript has been judged scientifically suitable for publication and will be formally accepted for publication once it meets all outstanding technical requirements.

Kind regards,

Engelbert A. Nonterah, MD, PhD

Academic Editor

PLOS ONE
---

## [Editor Report · Acceptance letter]

27 Sep 2024

PONE-D-24-08911R1 

PLOS ONE

Dear Dr. Ziauddeen, 

I'm pleased to inform you that your manuscript has been deemed suitable for publication in PLOS ONE. Congratulations! Your manuscript is now being handed over to our production team.

Kind regards, 

on behalf of

Dr. Engelbert Adamwaba Nonterah 

Academic Editor

PLOS ONE